# Oral health in Brazil: What were the dental procedures performed in Primary Health Care?

Maria Tereza Abreu Scalzo[1], Mauro Henrique Nogueira Guimarães Abreu[2], Antônio Thomaz Gonzaga Matta-Machado[3], Renata Castro Martins[3] *

1 School of Dentistry, Universidade Federal de Minas Gerais, Belo Horizonte, Minas Gerais, Brazil,
2 Department of Community and Preventive Dentistry, School of Dentistry, Universidade Federal de Minas Gerais, Belo Horizonte, Minas Gerais, Brazil, 3 Department of Preventive and Social Medicine, School of Medicine, Universidade Federal de Minas Gerais, Belo Horizonte, Minas Gerais, Brazil

☯ These authors contributed equally to this work.
* rcmartins05@gmail.com

**Data Availability Statement:** All relevant data are within the paper and its Supporting Information files.

## Abstract

This cross-sectional study aims to describe the primary dental care procedures performed by Oral Health Teams (OHTs), adhering to the third cycle of the "National Program for Improving Access and Quality of Primary Care" (PMAQ-AB) in Brazil. A descriptive analysis was performed through 26 dental procedures, including spontaneous, preventive, restorative/prosthetic and surgical procedures, and actions of cancer monitoring. Each conducted procedure assigned a score to the OHT, the final score being the sum of the number of procedures performed by the OHTs. These scores were then compared among the geographic regions of the country. Most OHTs perform basic dental procedures, such as supragingival scaling, root planning and coronal polishing (98.1%), composite filling (99.0%), and permanent tooth extraction (98.6%). The frequency related to dental prosthesis and monitoring of oral cancer decreased. Only 12.9% of the OHTs carries out biopsies, 30.9% monitor patients undergoing biopsy, 15.1% carry out impression for prostheses, and 13.6% carry out prostheses' installation. The scores reveal that OHT's performed, on average, 19.45 (±3.16) dental procedures. The OHTs in the South, Southeast, and Northeast had a higher number of primary dental procedures, while the teams in the North and Midwest performed, on average, fewer procedures. The Brazilian regions with the highest dental need have the lowest number of dental procedures. It is necessary to increase the range of procedures offered by OHT and reduce regional inequalities, adapting to the needs of the population in order to achieve comprehensive oral health.

## Introduction

Primary Health Care (PHC) in Brazil is considered the entry point for the user of the Brazilian National Health System (SUS in Portuguese). The evaluation of this level is strategic to

**Funding:** The author Mauro Henrique Nogueira Guimarães de Abreu received funding for this work from Fundação de Amparo à Pesquisa do Estado de Minas Gerais – FAPEMIG N° PPM 00148-17.

**Competing interests:** The authors have declared that no competing interests exist.

identifying the persistent fragilities that hinder its organization and operation towards the desired resolubility for the service [1].

To reorganize oral health actions in primary health care, the Brazilian Ministry of Health proposed the inclusion of Oral Health Teams (OHT) in the Family Health Strategy (FSH). FHS is the starting point for the application of the principles of PHC in Brazilian oral health. The result was the expansion of access to dental care in Brazil, [2,3] with improved dental facilities in the units and a better qualification of professionals [4]. In Brazil, there are currently 27,564 OHTs distributed throughout 5,032 municipalities [5].

The expansion of access to oral health has revealed the need to evaluate services in order to improve the quality of care provided to the population. Issues related to the quality of management and team practices assumed a more relevant position while it reduced the concern only with the expansion of services [6].

In 2011, the Brazilian Ministry of Health (MofH) launched the National Program for Improving Access and Quality of Primary Care (PMAQ-AB in Portuguese) [7]. The PMAQ-AB has the aim to increase access and improve the quality of health care, providing a national, regional, and local quality standard. This program is the largest health service evaluation program ever instituted in the country [8].

The PMAQ-AB is organized in three phases; adhesion and agreement, certification and re-agreement, and one strategic transverse axis of development [9], forming continuous evaluation cycles. In 2018, the 3rd cycle was finished. A significant increase was observed in the participation of OHTs over the years in PMAQ-AB. In the 1st (2011–2012), 2nd (2013–2014), and 3rd (2017–2018) cycles, the number of OHTs assessed were 12,403; 18,114, and 25,090, respectively.

After three cycles of PMAQ-AB, and approximately seven years of evaluation of dental services, studies have evaluated the procedures performed at PHC by OHTs. Reis et al. [10] and Neves et al. [11], using data from the 1st cycle, showed that most OHTs performed emergency, preventive, restorative and surgical procedures; however, for oral cancer and prosthetic procedures, a failure was found in both supply and execution. In the 2nd cycle, an improvement was observed in the external evaluation instrument with a greater number of variables related to the structure and work process [12]. However, Mendes et al. [13] revealed that failures related to the prevention and detection of oral cancer and the manufacture of prostheses still persisted. The evaluation of the 3rd cycle may indicate if there were advances in the provision of PHC procedures in oral health and changes in the performance of the OHTs, providing data for the improvement and development of services.

Brazil has a territory with a continental dimension and is divided into five regions: South, Southeast, Midwest, Northeast and North. This territorial extension leads to geographical socioeconomic inequalities with impact on health services and consequently on the population´s health [14].

The present study aims to describe the procedures of primary dental health care performed by OHTs that adhered to the 3rd cycle of the PMAQ-AB among Brazilian regions. The null hypothesis of this study is that there is no difference in the PHC procedures performed by OHT's among the five Brazilian regions.

## Methods

This study was approved by the Research Ethics Committee of Universidade Federal de Minas Gerais (UFMG), logged under protocol number 02396512.8.0000.5149. No participant was identified at any stage of this study, as it dealt with secondary and public data of the Brazilian MofH. This is a cross-sectional descriptive study that used secondary data from the third cycle of the PMAQ-AB, related to the procedures performed by OHTs that complied with the program.

In the 3<sup>rd</sup> cycle of PMAQ-AB, which occurred between 2017 and 2018, 25,090 OHTs were assessed. Of these, 2,097 (8.4%) were disqualified by the PMAQ-AB evaluation criteria, as they did not follow the program's recommendations, such as an adequate oral health surveillance system and the presence of the dentist and dental equipment in the PHC unit, resulting in a sample of 22,993 OHTs.

The data were collected during the external evaluation phase, which is characterized by a visit of program evaluators to the PHC units. An instrument, developed for this purpose by MofH in partnership with Brazilian teaching and research institutions, was used to interview the dental professionals and was applied by a trained team. At the time of the interview, the documents were verified to check the quality standards established according to norms, protocols, principles, and guidelines for the organization of actions and practices. The answers were recorded on tablets, using a program developed for PMAQ-AB. After data collection, the partner institutions performed the data validation and sent the results to the MofH's central database.

For this study, data were obtained from the interview conducted with the OHT and verification of documents in the PHC, through Module VI—Oral Health Professional Interview; section VI.7 - Organization of team's agenda and actions offered and section VI.11 - oral cancer care. Questions about 26 procedures performed by OHTs, mostly dichotomous (yes/no), were evaluated. The questions included response to spontaneous demand (subsection VI.7.4); preventive, restorative, surgical and procedures related to the manufacture and installation of dental prostheses (subsection VI.7.5); and actions of cancer monitoring (subsections VI.11.1, VI.11.2, VI.11.3, VI.11.6, VI.11.6/1, VI.11.7, VI.11.9).

Each procedure performed attributed a score to each OHT, with the final score being the sum of the quantity of procedures performed by each OHT (from 0 to 26 points). For example, if an OHT score was 18, it meant that this OHT carries out 18 of the 26 procedures evaluated. After this, the OHTs were divided into five geographic regions (North, Northeast, Midwest, South, and Southeast) in Brazil, and the score for each region was calculated.

The data were analyzed descriptively, by frequency, using the Statistical Package for Social Sciences (SPSS), version 25.0 (IBM SPSS Statistics for Windows, Armonk, NY).

## Results

The descriptive analysis showed that 98% of OHTs guarantee the scheduling of appointments and the spontaneous meeting of demand. Table 1 shows the organization of the spontaneous demand and the frequency of 26 primary dental care procedures performed by the OHTs, including preventive, restorative/prosthetic, surgical and actions of cancer monitoring. Most of OHTs perform basic procedures, such as supragingival scaling, root planning, and coronal polishing (98.1%); composite fillings (99.0%), and permanent tooth extractions (98.6%). However, it was observed that the frequency decreased when asked about procedures related to dental prostheses. Only 15.1% of the OHTs perform impressions for prostheses and 13.6% conduct prostheses' installations. With regard to actions for cancer prevention, most of the OHTs perform actions to prevent and diagnose oral cancer (94.9%), including medical advice on tobacco use (93.2%) as well as on alcohol and other drugs (91.7%). This percentage decreased slightly when ask about the active search for potentially precancerous lesions and cases in the population (80.5%). In addition, most OHTs did not perform biopsies (12.0%) and did not monitor patients undergoing biopsy to evaluate the results of the examination (30.9%).

Table 2 shows the scores of basic procedures that OHT's performed, which was, on average, 19.45 (±3.16) dental procedures (0–26 procedures). Regarding the average number of procedures performed according to the Brazilian regions, it was observed that the OHTs in the

**Table 1. Basic dental procedures performed by oral health teams, Brazil, 2017–2018.**

| Variables | Yes (%) |
|---|---|
| **Response to spontaneous demand** | |
| Urgency care (drainage of abscess, suture of trauma injuries, access to dental pulp, pulpotomy, treatment of alveolitis, initial treatment of traumatized tooth) | 22,298 (97.0) |
| Prescription of medicines | 22,360 (97.2) |
| Medical advice | 22,253 (96.8) |
| **Preventive and periodontal procedures** | |
| Fluoride application | 22,651 (98.5) |
| Ionomer sealant application | 20,102 (87.4) |
| Supragingival scaling, root planning, and coronal polishing | 22,566 (98.1) |
| Subgingival scaling, root planning, and coronal polishing | 20,117 (87.5) |
| **Restorative/Prosthetic procedures** | |
| Composite filling | 22,767 (99.0) |
| Amalgam filling | 18,450 (80.2) |
| Ionomer filling | 22,310 (97.0) |
| Impression for prostheses | 3,471 (15.1) |
| Prostheses installation | 3,114 (13.6) |
| Prostheses cementation | 5,785 (25.2) |
| **Surgical procedures** | |
| Deciduous tooth extraction | 22,812 (99.2) |
| Permanent tooth extraction | 22,680 (98.6) |
| Ulotomy/ulectomy | 18,803 (81.8) |
| **Cancer monitoring** | |
| Does the OHT carry out actions to prevent and diagnose oral cancer? | 21,817 (94.9) |
| Medical advice on tobacco use | 21,439 (93.2) |
| Medical advice on the use of alcohol and other drugs | 21,076 (91.7) |
| Medical advice on the prevention of exposure to solar radiation | 19,929 (86.7) |
| Active search for potentially precancerous lesions and cases in the population | 18,498 (80.5) |
| Systematic examination of oral mucosa | 20,468 (89.0) |
| Does the OHT perform biopsies to diagnose oral cancer? | 2,769 (12.0) |
| Does the OHT have a record of patients with suspected oral cancer who were biopsied or referred to the referral service? | 7,831 (34.1) |
| Does the OHT have a document that proves this? | 6,891 (30.0) |
| Does the OHT monitor patients undergoing biopsy to evaluate the results of the examination? | 7,109 (30.9) |
| After the user's reference for treatment, does the OHT follow up and monitor the continuity of care? | 14,645 (63.7) |

**Table 2. Scores of dental procedures performed by OHTs in Brazilian geographic regions, Brazil, 2017–2018.**

| Brazilian Regions | OHT | % | Mean | Minimum | Maximum | Median |
|---|---|---|---|---|---|---|
| North | 1,916 | 7.6 | 16.91 | 2 | 25 | 18 |
| Northeast | 11,132 | 44.4 | 19.46 | 0 | 26 | 20 |
| Midwest | 2,026 | 8.1 | 18.8 | 2 | 26 | 19 |
| Southeast | 6,751 | 26.9 | 20.16 | 0 | 26 | 20 |
| South | 3,265 | 13.0 | 19.84 | 3 | 26 | 20 |

South, Southeast, and Northeast had a higher number of primary dental procedures, while the teams in the North and Midwest, on average, performed fewer procedures.

## Discussion

The descriptive analysis revealed that most OHTs perform basic dental procedures, as recommended by the MofH, including emergency, preventive, restorative and surgical procedures. Regarding dental prostheses, few teams perform impressions, installations, and cementation of dental prostheses at the PHC. Although most teams perform actions to prevent and diagnose oral cancer, few teams perform biopsies of suspected cases, keep adequate records, or monitor patients undergoing biopsy to evaluate the results of the examination. In addition, the results revealed differences between Brazilian regions in relation to the procedures performed by the teams, invalidating the null hypothesis.

Despite changes in data collection methodology in the 3rd cycle, our findings are similar to previous 1st cycle [10,11] and 2nd cycle [13,15,16] studies. The highest perform of basic oral health procedures by OHTs, like emergency, preventive, restorative and surgical procedures, can be explained by the improvement in the infrastructure of dental offices and working conditions, and higher qualification of professionals due to the increase in investments in oral health, provided by the National Oral Health Policy [17]. The work process of oral health care is affected by legislation of the country [18], and access to care is influenced by the social determinants of health [19]. The delivery of health promotion strategies at the population level has shown a great impact on reducing the prevalence of oral diseases [19,20]. Health systems that strength preventive and tooth-preserving strategies, inclusive of adults, progress faster and perform better in respect of effectiveness and efficiency [20]. Over the past several decades, Netherlands passed by changes in the funding of oral health care aiming to reduce the need for curative treatment and more emphasis on prevention of dental diseases. These changes improved oral health, especially, Dutch adults [18]. Although progress has been made in the offer and organization of services, access has been expanded and oral health actions have been qualified, considering that, 17 years after the launch of the National Oral Health Policy, there are still difficulties to implement basic premises, such as completeness and access to secondary care and reduction of regional inequalities [21]. So, more political actions are needed to reduce inequalities, to promote health [19], and to improve the work process of OHTs.

Our results showed that most OHTs provide spontaneous demand care, together with emergency care, prescription of medication, or specialized guidance. Despite the advances in oral health services in recent years [17], the population still has a repressed demand for dental treatment, which leads to the search for emergency care [22]. Considering the current moment of the COVID-19 pandemic, when most routine dental care is not available, an increase in repressed demand is expected, causing more patients to seek out emergency dental care [23]. In this sense, it is expected that OHTs will be able to meet the demand, since they claim to be performing these types of procedures. In this sense, the emergency dental care must be

performed with a minimal use of slow and high-speed handpieces in order to avoid aerosols, prioritizing the use of manual instruments. In addition, the team must also have all personal protective equipment 's recommended and indicated by the relevant health authorities [24], demonstrating the importance of also having a well-structured service in terms of instruments and inputs.

Regarding preventive and periodontal procedures, it was observed that OHTs perform sub-gingival scaling less frequently. It is important to stress that this procedure, together with the correct diagnosis and oral hygiene instructions, is recommended as a periodontal preventive treatment, which is able to solve most demands, thus avoiding referral to secondary care [25].

It is important to note that the prevalence of tooth loss, which leads to the need for prosthetic rehabilitation, is still high in Brazil [26]. The findings on the manufacture and installation of dental prostheses show the failure in this process, although OHTs are required to offer this type of service. Reis et al. [10] Mendes et al. [13] and Cunha et al. [15] analyzing data of 1$^{st}$ and 2$^{nd}$ cycles of PMAQ-AB, also reveal failures in the procedures related to the offer and manufacture of dental prostheses. This demonstrates that there was no significant progress in this period. Prosthesis procedures require infrastructure and qualified labor for the service. The availability and distribution of regional prosthesis laboratories has not followed the epidemiological need of the population, and the growth of prosthesis production has been discrete in recent years [16]. This failure reveals a gap in the completeness of care, as it does not provide the complete treatment necessary to meet the user needs [21]. Besides that, the organizational factors and human resources has an essential role of in the provision of dental prosthesis in primary dental care in Brazil. The OHTs that are more likely to perform dental prostheses have professionals admitted through public examinations and involved in permanent education, with a more organized work process and that receive more support from municipal management [27].

Regarding oral cancer actions, the results reveal that, although oral cancer OHTs report procedures for prevention and early detection, few teams perform lesion biopsies. Results similar were founded by Galante et al. [28] using data of second cycle of PMAQ-AB. The presence of the dentist in primary care, had a positive impact on campaign actions, follow-up, referral to specialists, and registration of suspected cases of oral cancer However, actions to perform lesion biopsies are done on secondary care. This is a worrisome fact due to the high mortality rate of the disease [29]. The active search for lesions, visual examination for oral cancers during regular dental consultations, and biopsy of suspected lesions may be able to reduce mortality from oral cancer [30], mainly when these actions are directed toward alcohol and tobacco users, since these are the main risk factors for oral cancer, which is linked to more than 80% of the cases [31]. Early diagnosis is crucial for patient survival, since survival rates for oral cancer are associated with the stage of the tumor in diagnosis, as well as the availability and quality of treatment provided [32]. In addition, the knowledge and experience of health professionals also plays a key role; monitoring post-treatment patients prevents recurrence and improves patients' quality of life [33], since the treatment of an oral cancer does not necessarily mean a cure. Nevertheless, our results revealed that OHTs do not continuously monitor oral cancer cases. Due to its structure and organization, the PHC is an excellent strategy for screening oral cancer cases. The access of community agents to patients' homes can help in early detection and follow-up of suspected and confirmed cases [10]. Therefore, it is important to qualify professionals and encourage campaigns for systematic examinations of oral mucosa as a way to qualify the diagnosis of potentially malignant lesions in a more frequent manner than that reported in our findings.

Regarding the scores obtained in this study, better rates were observed in the Southeast and South regions, followed by the Northeast region. The North region presented the worst results,

showing important differences between the regions. This difference can be explained by the level of development of these regions. Despite the growth of the Municipal Human Development Index (HDI), which has proven to be more accelerated in recent years in Brazil, with improvement in the indices of all regions, the North and Northeast still have the lowest HDI in the country. The highest HDI is recorded in the Southeast region (0.766), followed by the Midwest (0.757) and South (0.754) regions [34]. From 2004, with National Oral Health Policy, an increase in the oral health incentive was observed for municipalities with lower HDIs. This policy benefited the North and Northeast regions, with increased access and improved infrastructure. However, the impact of this policy on service use was not enough, since the proportion of procedures is still higher among regions with higher HDIs [28,35].

In recent years, Brazil has expanded oral health service coverage and there have been changes in the epidemiological profile of oral diseases, but efforts are still needed to reduce inequalities in access to services, in improved and qualified care, as well as in the use epidemiology for planning oral health actions [11,19]. The Northeast region has a higher number of OHTs in the country, while the North region presents the worst HDI of the country and the lowest number of OHTs. However, this number does not reflect a higher score of dental procedures performed by OHTs when compared to the Southeast and South regions.

Therefore, the results showed that only the number of OHTs does not necessarily translate into better care for the population. Other factors, such as different socio-demographic conditions, repressed demand for dental treatment, and the organization of the dental service may interfere in the offer of the service to the population.

This study's results are limited due to the use of a secondary database, based on dentist reports, and by the fact that PMAQ-AB is a pay-for-performance program. Nevertheless, the need to improve and expand the supply of prostheses and actions related to the early diagnosis and monitoring of oral cancer is evident. Despite these limitations, our study is based on national data, from the largest PHC evaluation program ever conducted in Brazil in its third cycle [8], which reinforces the relevance of our results.

It is necessary to increase the range of procedures offered by OHTs and reduce regional inequalities, adapting them to the needs of the population in order to achieve comprehensive oral health care. Health assessments can provide an overview that allows one to trace ways to boost access to oral health.

## Supporting information

**S1 Dataset.**
(XLSX)

## Acknowledgments

Coordenação de Aperfeiçoamento de Pessoal de Nível Superior (CAPES 001), Conselho Nacional de Desenvolvimento Científico e Tecnológico (CNPq), Fundação de Amparo à Pesquisa do Estado de Minas Gerais (FAPEMIG), and Pró-Reitoria de Pesquisa da Universidade Federal de Minas Gerais (PRPq-UFMG).

## Author Contributions

**Conceptualization:** Mauro Henrique Nogueira Guimarães Abreu, Renata Castro Martins.

**Data curation:** Antônio Thomaz Gonzaga Matta-Machado.

**Formal analysis:** Maria Tereza Abreu Scalzo, Mauro Henrique Nogueira Guimarães Abreu, Renata Castro Martins.

**Investigation:** Maria Tereza Abreu Scalzo.

**Methodology:** Mauro Henrique Nogueira Guimarães Abreu, Renata Castro Martins.

**Project administration:** Renata Castro Martins.

**Supervision:** Renata Castro Martins.

**Writing – original draft:** Maria Tereza Abreu Scalzo.

**Writing – review & editing:** Mauro Henrique Nogueira Guimarães Abreu, Antônio Thomaz Gonzaga Matta-Machado, Renata Castro Martins.

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
