## [Decision Letter · Decision Letter 0]

11 Aug 2021

PONE-D-21-14792

Oral health in Brazil: what were the dental procedures performed in Primary Health Care?

PLOS ONE

Dear Dr. Martins,

Thank you for submitting your manuscript to PLOS ONE. After careful consideration, we feel that it has merit but does not fully meet PLOS ONE’s publication criteria as it currently stands. Therefore, we invite you to submit a revised version of the manuscript that addresses the points raised during the review process.

We look forward to receiving your revised manuscript.

Kind regards,

Gaetano Isola, Ph.D.

Academic Editor

PLOS ONE

1. Please ensure that your manuscript meets PLOS ONE's style requirements, including those for file naming. The PLOS ONE style templates can be found at https://journals.plos.org/plosone/s/file?id=wjVg/PLOSOne_formatting_sample_main_body.pdf and https://journals.plos.org/plosone/s/file?id=ba62/PLOSOne_formatting_sample_title_authors_affiliations.pdf.

2. Acknowledgments Section: Move New Information to the Financial Disclosure:

"Thank you for stating the following in the Acknowledgments Section of your manuscript:

“This study was supported by the Coordenação de Aperfeiçoamento de Pessoal de Nível Superior (CAPES), Conselho Nacional de Desenvolvimento Científico e Tecnológico (CNPq), and Pró-Reitoria de Pesquisa da Universidade Federal de Minas Gerais (PRPq-UFMG).”

Additional Editor Comments (if provided):

Reviewers' comments:

Reviewer's Responses to Questions

**Comments to the Author**

1. Is the manuscript technically sound, and do the data support the conclusions?

Reviewer #1: Partly

Reviewer #2: Yes

2. Has the statistical analysis been performed appropriately and rigorously? 

Reviewer #1: N/A

Reviewer #2: Yes

3. Have the authors made all data underlying the findings in their manuscript fully available?

Reviewer #1: Yes

Reviewer #2: Yes

4. Is the manuscript presented in an intelligible fashion and written in standard English?

Reviewer #1: Yes

Reviewer #2: Yes

5. Review Comments to the Author

Reviewer #1: The data were very old (2017-2018) and These data in present study is not contemporary, so these results don't reflect the current oral health requirements in Brazil.

Material method

Explain 5 macro region, mention which city in five region.

Reviewer #2: Thank you for submitting you manuscript.

The manuscript presents good data about Brazilian national public program for oral health services. Public health professionals can use these valuable data in countries where they provide dental public health services.

Manuscript is organized and present sound interpretation of data, but needs some language editing and review to avoid confusion and make it more clear and readable. Especially for some terminologies that have been translated to English terms.

Here are some examples (in bold and underline):

69 three phases: accession and contractualisation, certification and recontratualisation, and a transversal strategic axis of development

78 OHTs performed individual preventive, restorative, and surgical procedures and

emergencies; however

132 performed by the OHTs, including preventive, restorative, surgical, and prosthetic

133 procedures and cancer monitoring.

142 and cases in the community (80.5%). In addition, most OHTs did not realize biopsies

Also in Discussion:

The great accomplishment of basic oral health procedures by OHTs can be explained by the improvement in the infrastructure of dental offices, improvement in working conditions and higher qualification of professionals due to the increase in investments in oral health, provided by the National Oral Health Policy [16].

Comment: if you can elaborate more and exaplain what do you mean by the great accomplishment? Is it improvement in provided services compared to the 2nd cycle? Maybe mention some data from 2nd cycle to make it clear.

In this sense, the emergency dental care must be performed with a minimal use of high

speed handpieces in order to avoid aerosols resulting from the use of manual instruments.

Comment : you mean headpiece instrument?

6. PLOS authors have the option to publish the peer review history of their article (what does this mean?). If published, this will include your full peer review and any attached files.

Reviewer #1: No

Reviewer #2: No

---

## [Author Response · Author response to Decision Letter 0]

24 Sep 2021

We would like to thank the reviewers for the excellent suggestions in order to improve our paper. The responses of each comment are listed below. All edition and revisions made are highlighted by red in the new file. 

Reviewer #1:

The data were very old (2017-2018) and these data in present study is not contemporary, so these results don't reflect the current oral health requirements in Brazil.

Response: The National Program for Improving Access and Quality of Primary Health Care (PMAQ-AB) is the largest health service evaluation program instituted in Brazil. The program ended in 2018 after three evaluation cycles. This study is based on national data, from the largest PHC evaluation program ever conducted in Brazil. These data refer to the last cycle of the PMAQ. Until this moment, there is no health service evaluation program under development in the country. So, nowadays, these are the data available that represents Brazilian national public program for oral health services, which reinforces the relevance of these results. 

Material method

Explain 5 macro region, mention which city in five region.

Response: Thanks for the observations. We agree with the reviewer and the explanation about the Brazilian regions was included in Introduction section (7th paragraph, 84 line; The differences between the regions are discussed from 230 Line in Discussion. 

Reviewer #2:

Thank you for submitting you manuscript. The manuscript presents good data about Brazilian national public program for oral health services. Public health professionals can use these valuable data in countries where they provide dental public health services.

Manuscript is organized and present sound interpretation of data, but needs some language editing and review to avoid confusion and make it more clear and readable. Especially for some terminologies that have been translated to English terms.

Here are some examples (in bold and underline):

69 three phases: accession and contractualisation, certification and recontratualisation, and a transversal strategic axis of development

Response: Thanks for the observations. We agree with the reviewer. The text was rewritten and the change was made to “The PMAQ-AB is organized in three phases; adhesion and agreement, certification and re-agreement, and one strategic transverse axis of development...”

78 OHTs performed individual preventive, restorative, and surgical procedures and emergencies; however

Response: Thanks for the observations. We agree with the reviewer. The text was rewritten and the change was made to “OHTs performed emergency, preventive, restorative and surgical procedures; however…”

132 performed by the OHTs, including preventive, restorative, surgical, and prosthetic 133 procedures and cancer monitoring.

Response: Thanks for the observations. We agree with the reviewer. The text was rewritten and the change was made to “performed by the OHTs, including preventive, restorative/prosthetic, surgical and actions of cancer monitoring.”

142 and cases in the community (80.5%). In addition, most OHTs did not realize biopsies

Response: Thanks for the observations. We agree with the reviewer. The text was rewritten and the change was made to “and cases in the population (80.5%). In addition, most OHTs did not perform biopsies”

Also in Discussion:

The great accomplishment of basic oral health procedures by OHTs can be explained by the improvement in the infrastructure of dental offices, improvement in working conditions and higher qualification of professionals due to the increase in investments in oral health, provided by the National Oral Health Policy [16].

Comment: if you can elaborate more and exaplain what do you mean by the great accomplishment? Is it improvement in provided services compared to the 2nd cycle? Maybe mention some data from 2nd cycle to make it clear

Response: Thanks for the observations and the opportunity to clarify this point. The better performance of OHTs, in basic procedures, is due to a greater investment in oral health, after the National Oral Health Policy. The text was rewritten: “The highest perform of basic oral health procedures by OHTs, like emergency, preventive, restorative and surgical procedures, can be explained by the improvement in the infrastructure of dental offices and working conditions, and higher qualification of professionals due to the increase in investments in oral health, provided by the National Oral Health Policy [17].”

In this sense, the emergency dental care must be performed with a minimal use of high speed handpieces in order to avoid aerosols resulting from the use of manual instruments. Comment: you mean headpiece instrument?

Response: Thanks for the observations and the opportunity to clarify this point. The text was rewritten:” In this sense, the emergency dental care must be performed with a minimal use of slow and high-speed handpieces in order to avoid aerosols, prioritizing the use of manual instruments.”

---

## [Decision Letter · Decision Letter 1]

11 Nov 2021

PONE-D-21-14792R1

Oral health in Brazil: what were the dental procedures performed in Primary Health Care?

PLOS ONE

Dear Dr. Martins,

Thank you for submitting your manuscript to PLOS ONE. After careful consideration, we feel that it has merit but does not fully meet PLOS ONE’s publication criteria as it currently stands. Therefore, we invite you to submit a revised version of the manuscript that addresses the points raised during the review process.

We look forward to receiving your revised manuscript.

Kind regards,

Gaetano Isola, Ph.D.

Academic Editor

PLOS ONE

Journal Requirements:

Additional Editor Comments:

Please revise in accordance to reviewer's 3 comments before any further assessment of the manuscript.

Reviewers' comments:

Reviewer's Responses to Questions

**Comments to the Author**

1. If the authors have adequately addressed your comments raised in a previous round of review and you feel that this manuscript is now acceptable for publication, you may indicate that here to bypass the “Comments to the Author” section, enter your conflict of interest statement in the “Confidential to Editor” section, and submit your "Accept" recommendation.

Reviewer #2: All comments have been addressed

Reviewer #3: All comments have been addressed

2. Is the manuscript technically sound, and do the data support the conclusions?

Reviewer #2: Yes

Reviewer #3: Yes

3. Has the statistical analysis been performed appropriately and rigorously? 

Reviewer #2: Yes

Reviewer #3: I Don't Know

4. Have the authors made all data underlying the findings in their manuscript fully available?

Reviewer #2: Yes

Reviewer #3: Yes

5. Is the manuscript presented in an intelligible fashion and written in standard English?

Reviewer #2: Yes

Reviewer #3: Yes

6. Review Comments to the Author

Reviewer #2: (No Response)

Reviewer #3: This is a very interesting article.

I would suggest the authors to search if other similar articles were performed in other countries.

In my opinion the materials and methods chapter has to be better explained.

Please also in the discussion chapter try to find more recent published articles.

7. PLOS authors have the option to publish the peer review history of their article (what does this mean?). If published, this will include your full peer review and any attached files.

Reviewer #2: **Yes: **Ziyad Allahem

Reviewer #3: No

---

## [Author Response · Author response to Decision Letter 1]

22 Dec 2021

We would like to thank the reviewers for the excellent suggestions in order to improve our paper. The responses of each comment are listed below. All edition and revisions made are highlighted by red in the new file. 

Additional Editor Comments

Please revise in accordance to reviewer's 3 comments before any further assessment of the manuscript.

6. Review Comments to the Author

Reviewer #2: (No Response)

Reviewer #3:This is a very interesting article. would suggest the authors to search if other similar articles were performed in other countries. In my opinion the materials and methods chapter has to be better explained. Please also in the discussion chapter try to find more recent published articles.

Response: Thanks for the observations. There is no program for evaluation of quality of primary care similar to PMAQ, in other countries. However, we added three articles that discuss the systems of oral care in an international perspective (references 18, 19 and 20). The materials and methods chapter was reviewed to be more clear Recent published articles were added to. discussion chapter (references 18, 19, 20, 27 and 28).

---

## [Decision Letter · Decision Letter 2]

17 Jan 2022

Oral health in Brazil: what were the dental procedures performed in Primary Health Care?

PONE-D-21-14792R2

Dear Dr. Martins,

We’re pleased to inform you that your manuscript has been judged scientifically suitable for publication and will be formally accepted for publication once it meets all outstanding technical requirements.

Kind regards,

Gaetano Isola, Ph.D.

Academic Editor

PLOS ONE

Additional Editor Comments (optional):

The authors have well addressed to all comments raised by both reviewers, as clearly shown in the reviewer round #2 report. No further issues are needed.

Reviewers' comments:

Reviewer's Responses to Questions

**Comments to the Author**

Reviewer #2: All comments have been addressed

Reviewer #3: All comments have been addressed

2. Is the manuscript technically sound, and do the data support the conclusions?

Reviewer #2: Yes

Reviewer #3: Yes

3. Has the statistical analysis been performed appropriately and rigorously? 

Reviewer #2: N/A

Reviewer #3: Yes

4. Have the authors made all data underlying the findings in their manuscript fully available?

Reviewer #2: Yes

Reviewer #3: Yes

5. Is the manuscript presented in an intelligible fashion and written in standard English?

Reviewer #2: Yes

Reviewer #3: Yes

6. Review Comments to the Author

Reviewer #2: (No Response)

Reviewer #3: The authors have responded to all my demands and reviewed the article according to my indications.

7. PLOS authors have the option to publish the peer review history of their article (what does this mean?). If published, this will include your full peer review and any attached files.

Reviewer #2: **Yes: **Ziyad Allahem

Reviewer #3: No

---

## [Editor Report · Acceptance letter]

20 Jan 2022

PONE-D-21-14792R2 

Oral health in Brazil: what were the dental procedures performed in Primary Health Care?  

Dear Dr. Martins:

I'm pleased to inform you that your manuscript has been deemed suitable for publication in PLOS ONE. Congratulations! Your manuscript is now with our production department. 

Kind regards, 

on behalf of

Prof. Gaetano Isola 

Academic Editor

PLOS ONE